# Effects of Additional Granola in Children’s Breakfast on Nutritional Balance, Sleep and Defecation: An Open-Label Randomized Cross-Over Trial

**DOI:** 10.3390/children10050779

**Published:** 2023-04-25

**Authors:** Yuma Matsumoto, Hiroyuki Sasaki, Hirofumi Masutomi, Katsuyuki Ishihara, Shigenobu Shibata, Kazuko Hirao, Akiko Furutani

**Affiliations:** 1Research & Development Division, Calbee, Inc., 23-6 Kiyohara-Kogyodanchi, Utsunomiya, Tochigi 321-3231, Japan; y_matsumoto1@calbee.co.jp (Y.M.); k_ishihara@calbee.co.jp (K.I.); 2Laboratory of Physiology and Pharmacology, School of Advanced Science and Engineering, Waseda University, Shinjuku-ku, Tokyo 162-8480, Japan; hiroyuki-sasaki@aoni.waseda.jp (H.S.); shibatas@waseda.jp (S.S.); furutani@aikoku-jc.ac.jp (A.F.); 3Division of Home Economics, Aikoku Gakuen Junior College, Edogawa-ku, Tokyo 133-0057, Japan; hirao@aikoku-jc.ac.jp

**Keywords:** nutritional balance, children, breakfast, sleep, defecation

## Abstract

The contribution of breakfast to daily nutrient intake is low, particularly among children, at only about 20%, and it is difficult to determine whether children are receiving adequate nutrients at breakfast. Although alterations in breakfast content are considered to affect lifestyle habits such as sleep and defecation, there have been few intervention studies in children. The relationship between nutritional balance, dietary intake, and lifestyle habits in children remains unclear. We conducted an intervention study on elementary school children’s breakfasts and observed the effects of improving the nutritional balance of breakfast on sleep parameters and defecation status. An intervention study was conducted with 26 elementary school students in Tokyo. The study design was an open-label randomized cross-over trial. Subjects consumed their usual breakfast during the control period and a granola snack containing soy protein in addition to their usual breakfast during the intervention period. Questionnaires regarding breakfast, sleep, and bowel movements were administered during each period. Based on the answers to these questionnaires, we compared the nutritional sufficiency of macronutrients, vitamins, and minerals (29 in total), as well as changes in sleep parameters and defecation status. The additional consumption of granola snacks increased the breakfast intake of 15 nutrients. The changes were particularly significant for iron, vitamin B1, vitamin D, and dietary fiber. During the intervention, sleep duration was decreased and wake-up time became earlier. In terms of defecation, the intervention did not change stool characteristics, but the frequency of defecations per week increased on average by 1.2 per week. These results suggest that the nutritional balance and the amount of breakfast are linked to sleep and defecation and that improving breakfast content can lead to lifestyle improvements in children.

## 1. Introduction

Diet is an essential aspect of maintaining good health, and dietary habits and quality have a significant impact on the body. The importance of the breakfast meal has been described in various reports. Irregular habits, such as unbalanced nutrition and skipping meals, can have a negative impact on the body and increase the risk of metabolic disorders, including type 2 diabetes and obesity [1,2,3]. The breakfast-skipping rate for 7–14 year-olds in Japan is 4.4% by the survey of the 2019 National Health and Nutrition Survey [4]. Furthermore, it is also becoming clear that children who skip breakfast tend to have poorer academic performance [5], and that alterations in breakfast content can affect lifestyle habits such as sleep and defecation [6,7]. In addition, the relationship between eating habits and the circadian clock has gradually become clearer in recent years and has attracted increasing attention.

Mammals have circadian rhythms of approximately 24 h per day, including sleep–wake and body temperature rhythms. The circadian rhythm is established via fluctuations in the expression levels of clock genes, and this system exists in a variety of tissues throughout the body, including the brain, liver, and muscles. The suprachiasmatic nucleus (SCA) of the brain serves as the central clock and controls the peripheral clocks of other tissues [8,9,10]. While the central clock is tuned by light stimulation, the peripheral clock is tuned by diet, in addition to central control. In other words, diet is also important in determining the clock of the whole body [11,12]. Given that each tissue has its own circadian rhythm, it is known that there are day–night differences in digestive and metabolic processes, and that breakfast content has a significant effect on the body clock [13]. It has been reported that raising the blood glucose level at breakfast promotes insulin secretion and synchronizes the body clock, but sugars such as glucose, sucrose, and fructose alone are not ideal; a well-balanced diet rich in protein is effective in synchronizing the body clock [14]. Furthermore, a hearty breakfast after a long fast is effective for entrainment of the body clock, suggesting that both balance and volume of nutritional intake are important [15].

Eating habits in Japan generally consist of three meals a day: breakfast, lunch, and dinner. However, most people do not consume their required energy and nutrients equally divided into three portions. The contribution from breakfast tends to be low, at about 20% to 25% of the total daily intake [16]. For example, the average energy intake at breakfast is 374 kcal for children (6–11 years old), 464 kcal for adolescents (12–17 years old), 411 kcal for young adults (18–49 years old), and 472 kcal for older adults (over 50 years old), or 20.8%, 20.4%, 21.2%, and 24.8% of the daily intake, respectively. This tendency toward breakfast deficiency is more pronounced in younger age groups, especially among adolescents. The breakfasts of young people have also been described as having lower contents of SFAs, MUFAs, n−3 PUFAs, dietary fiber, vitamins (vitamins A, E, K, B_6_, C, thiamin, and niacin), and potassium. It has been reported that the content and nutritional intake of breakfast is positively correlated with overall daily nutritional intake [17], and we consider breakfast inadequacy to be an issue that needs to be improved, especially for the physical and mental development of young people. However, preparing a nutritionally balanced breakfast can be a major burden.

With the globalization of diet in modern Japan, increasing numbers of people eat bread or cereals as staple foods at breakfast. Cereals have been widely accepted because they are easily prepared. In addition, a systematic review found that people who regularly consume breakfast cereals have higher daily intakes of dietary fiber, B group vitamins, folate, calcium, iron, magnesium, and zinc [18]. Therefore, cereal consumption for breakfast is considered useful from a nutritional point of view.

The physical benefits of eating cereal for breakfast have been reported in other studies [19], and we have previously reported the effects of replacing breakfast cereal on defecation and quality of life in adults [6]. However, reports of intervention studies in younger age groups are scarce. Here, we hypothesized that improving the nutritional balance of breakfast would change sleep and defecation outcomes in children; we conducted an intervention study in which Japanese elementary school children aged 6 to 12 added a granola snack containing soy protein to their usual breakfast, and observed its effects on their lifestyle and habits.

## 2. Materials and Methods

### 2.1. Subjects

Twenty-six children aged between 6 and 12, attending elementary school in Tokyo, and who met the following eligibility criteria were included in the study.

(i)Consent obtained from their parents or guardians.(ii)Have no medical conditions for which they are currently receiving treatment.(iii)Not taking any medication under the direction of a physician.(iv)Not planning to receive medical treatment during the study period.(v)Have no food allergies to milk, wheat, almonds, soybeans, or gelatin.

This study was conducted in accordance with the Declaration of Helsinki. The purpose, content, and ethical considerations of the study were explained to the guardians of the subjects in writing, and informed consent was obtained. Prior to conducting this study, approval was obtained from the Ethical Review Committee at Aikoku Gakuen Junior College (Approval date: 7 June 2022, Approval number: 2019-R04). Age, height, and weight of subjects in Groups A and B were added to the Appendix A, and both height and weight were close to the average for Japanese of that age.

### 2.2. Test Food

Granola snacks containing soy protein (Furugra Bits, Calbee, Inc., Tokyo, Japan) were used in this study. The appearance and nutritional composition of the snacks are shown in Appendix A, respectively.

### 2.3. Study Design

The study was an open-label randomized cross-over trial, registered in the University Hospital Medical Information Network (UMIN) registry (UMIN study ID: UMIN000049363). The trial flow is shown in Figure 1. Subjects were randomly divided into two groups (Group A and B) at the beginning the study. The study consisted of one week in which subjects ate their usual breakfast and one week of eating the test food in addition to their usual breakfast, with a one-week washout period between the test periods (The first intervention period: 31 October–6 November 2022, washout: 7–13 November 2022, the second intervention period: 14–20 November 2022). As November 3rd was a holiday in Japan, sleep data on that day were not included in the analysis. We asked participates to continue their usual breakfast habits during both two test periods and washout period. During the trial, subjects recorded their daily breakfast, bedtime, and wake-up time, and completed a questionnaire. In addition, after each trial period, subjects completed a questionnaire about their defecation over the week.

### 2.4. Sleep Analysis

The average bedtime and wake-up time on weekdays and weekends (free days), average sleep duration, and median sleep time on weekends, corrected for weekday sleep debt (MSFsc), were calculated for each subject based on the daily bedtime and wake-up time reported in the questionnaire [20].

### 2.5. Defecation Analysis

The defecation questionnaire included questions on the frequency of defecation per week, the number of days with defecation, the Bristol Stool scale (BSS), and questions based on the Constipation Assessment Scale (CAS). The BSS was based on visual examination of the stools and scored on a scale from 1 (hard, long transit time through the digestive tract) to 7 (watery, fast transit time through the digestive tract) [21]. The CAS is a method developed by McMillan and Williams (1989) to evaluate constipation [22], and a modified version in Japanese and for children was used in this study [23]. It consists of eight questions, with three options for each question. Each question was scored on a scale of 0–2, with 0 being the best and 2 being the worst, for a maximum total of 16 points. Note that a score of 5 or more is considered to indicate a problematic constipation condition.

### 2.6. Nutritional Calculation

Nutritional calculations were performed by a registered dietitian with over 5 years of experience, based on the breakfast contents reported in the questionnaire responses. Due to the different ages and genders of the subjects, all nutrients were converted into adequacy ratios, which were calculated as 100% of one-third of the Japanese Dietary Reference Intakes (2020). The calculated nutrient profiles were: energy, protein, total fat, carbohydrate, saturated fatty acids, n−6 unsaturated fatty acids, n−3 unsaturated fatty acids, dietary fiber, salt, potassium, calcium, magnesium, phosphorus, iron, zinc, copper, manganese, vitamin A, D, E (α-tocopherol), K, B_1_, B_2_, B_6_, B_12_, C, niacin, folate, and pantothenic acid. For each nutrient, the average value was calculated for each individual and then compared for the control period and for the additional granola snack intake period (intervention period). The software used for nutrient value calculations was Healthy Maker Pro (Mushroom Soft, Okayama, Japan).

### 2.7. Statistical Analysis

The sample size was calculated as 34 using G*Power software (University of Dusseldorf, Germany), with an effect size of 0.5, α of 0.05, and power (1−β) of 0.8. Statistical processing was performed using GraphPad Prism^®^ (MDF, Ltd., Tokyo, Japan). Questionnaire items related to defecation were analyzed using Wilcoxon’s signed-rank test for noncontinuous data. Other continuous data were checked for normality using the D’Agostino and Pearson test. Data for nutrient sufficiency were analyzed using Wilcoxon’s signed-rank test because they were all non-normally distributed. The data for sleep were all normally distributed. Data for sleep duration, bedtime, and wake-up time were analyzed using a two-way repeated analysis of variance (ANOVA) with a day-of-week factor (weekdays or weekends) and an intervention factor (control or intervention). MSFsc was analyzed using corresponding *t*-tests. For all statistical analyses, the significance level was set at 5%.

## 3. Results

The analysis included 21 patients who met the eligibility criteria and completed the trial (exclusions: one patient who did not adhere to the protocol and four patients who had problems completing the questionnaire) (Figure 1). One of the participants had answered in the pre-questionnaire that he had a cereal eating habit. However, we confirmed that he did not consume cereal as usual breakfast during the study period.

### 3.1. Nutritional Balance Was Improved by Additional Granola Snacks

The nutritional balance of the children’s breakfast was assessed by adequacy rate. The adequacy rates for the three macronutrients (protein, fat, and carbohydrate) in the usual breakfast were 83.6%, 85.6%, and 72.2%, respectively. In addition, the adequacy rates for minerals and vitamins were particularly low. When granola snacks with soy protein were added to the usual breakfast, there was a significant increase in adequacy rates for 15 nutrients (energy (69.6% to 84.1%), protein (83.6% to 93.9%), fat (85.6% to 99.9%), carbohydrates (72.2% to 81.2%), potassium (80.1% to 93.0%), calcium (64.0% to 82.8%), phosphorus (70.2% to 82.9%), iron (56.2% to 94.7%), vitamin D (61.2% to 87.6%), vitamin B_1_ (60.9% to 93.6%), vitamin B_6_ (71.8% to 94.4%), vitamin B_12_ (83.0% to 95.4%), folate (83.2% to 100.3%), pantothenic acid (68.2% to 89.1%), vitamin C (58.7% to 70.9%), and dietary fiber (72.0% to 98.4%)) (Figure 2A–C, Appendix A). The largest change was in iron, followed by vitamin B1, vitamin D, and dietary fiber (Figure 2D). This indicated that the additional intake of a granola snack brought the adequacy rate closer to 100% and improved the nutritional balance of the breakfast. There was no statistically significant difference in the sufficiency rates of n−3, n−6 PUFA and SFA, while the sufficiency rate of total lipids increased. Granola contains MUFA, which may be responsible for the increase in total lipids. However, we did not assess the intake of MUFA in this study. 

### 3.2. Additional Granola Snacks Affected Sleep Rhythms

A two-way ANOVA was performed for each sleep parameter included in the questionnaire, with a day-of-week factor (weekdays or weekends) and an intervention factor (control or intervention). While no interactions were found in any of the parameters, average sleep duration was decreased and wake-up time became earlier with the effect of the intervention (seep duration: intervention, F_(1,20)_ = 5.15, *p* < 0.05, η^2^ = 0.012; wake-up time: intervention, F_(1,20)_ = 9.59, *p* < 0.01, η^2^ = 0.020) (Figure 3A,B). The difference in average sleep duration between the control and intervention periods was 6 min on weekdays and 13 min on weekends (Appendix A). The difference in average wake-up time was 11 min on weekdays and 15 min on weekends. For all sleep parameters, differences by the day-of-the-week effect were observed. On weekends, bedtime and wake-up time were later and the average sleep duration was longer (sleep duration: day-of-week, F_(1,20)_ = 12.35, *p* < 0.01, η^2^ = 0.059; bedtime: day-of-week, F_(1,20)_ = 7.34, *p* < 0.05, η^2^ = 0.012; wake-up time: day-of-week, F_(1,20)_ = 21.30, *p* < 0.001, η^2^ = 0.18) (Figure 3A–C). Although the children’s sleep rhythms were altered by the intervention, no significant changes were found in MSFsc (Figure 3D).

### 3.3. Additional Granola Snacks Increased Defecation Frequency

The frequency of defecation increased by 1.2 per week on average during the intervention period, from an average of 6.5 times in the control period to 7.7 times in the intervention period (Figure 4A, Appendix A). This result suggests that the additional intake of granola snacks at breakfast may encourage defecation in children. On the other hand, no significant differences were found in the number of days with defecation per week, stool characteristics, or constipation scale (Figure 4B–D). Subjects in this study were mostly in good condition without intervention, with an average BSS score of 3.4 indicating the good condition of stool quality. In addition, only one subject scored 5 or above on the CAS, indicating that most were not constipated and were regularly in a good defecation status group.

## 4. Discussion

Murakami et al. investigated the nutritional intake at breakfast among Japanese participants in 2018 and reported that the average energy intake at breakfast for 6–11 year olds was 374 kcal [16]. The average energy intake during the control period in this study was 387.4 kcal (Appendix A), which was similar to the value indicated in that report, so the subjects in this study were considered to be an average breakfast-consuming population.

The addition of granola snacks containing a soy protein to the children’s breakfasts increased the nutritional adequacy rate for 15 nutrients. There were large improvements in adequacy rates of iron, vitamins, and dietary fiber, and the nutritional balance of breakfast was considered to have improved. As not only the frequency but also the nutritional balance (quality of the meal) is considered to be related to a reduced risk of obesity and type 2 diabetes [1], we consider that a breakfast format similar to that used in this study may contribute to a healthy lifestyle for children, although there is a need to explore the relationship between breakfast diet quality and lifestyle in more detail. In the future, it will be necessary to conduct a long-term intervention lasting several months, and to observe the effects on physical condition as well as lifestyle.

Analysis of sleep status showed that the addition of granola snacks to breakfast resulted in an earlier wake-up time and shorter sleep duration. In all cases (weekend/weekday, with/without intervention), the average sleep duration was more than 520 min, which means that the subjects had enough sleep [24]. Therefore, we consider that the earlier wake-up time due to the additional granola snacks does not cause sleep insufficiency. The reasons for the changes in sleep parameters are unclear, but three main possibilities were considered. First, mealtime durations may have increased according to the amount of food consumed, which may have led to earlier wake-up times. In fact, some reports have shown a correlation between dietary complexity and wake-up time [25]. Second, it is possible that the quality of sleep changed as a result of the consumption of a nutritionally balanced breakfast. Based on reports of an interrelationship between nutritional intake, including vitamin intake, and sleep quality [26,27], the improved nutritional balance might have affected sleep quality, leading to earlier waking. In future experiments, we intend to consider the quality and quantity of sleep using the Pittsburgh Sleep Quality Index (PSQI) questionnaire to investigate the relationship between nutritional balance and sleep quality. Third, it is possible that participants’ circadian rhythms changed. Hirao et al. reported that a balanced diet rich in protein and carbohydrates may help to reset the peripheral circadian clock [10]. As the nutritional adequacy rate of protein in this study increased from 83.6% to 93.9% (13.3 g to 17.1 g), it is possible that the improved nutritional balance affected the circadian clock. It has been reported that diet-induced secretion of insulin, IGF-1, and short-chain fatty acids (SCFAs) produced by dietary fiber is associated with a clock gene synchronization mechanism [11,28,29]. In this study, the addition of granola snacks increased nutrient adequacy rates for carbohydrates, protein, and dietary fiber, which stimulate the secretion of insulin, IGF-1, and SCFAs, respectively. It is possible that these may be involved in the synchronization of clock genes.

In this study, no changes were observed in MSFsc and social jet lag (Appendix A). Social jet lag was evaluated using sleep time differences between the weekdays and weekend. It is well known that social jet lag values are positively associated with not only school grades [30] but also with mental health scores such as depression, exhaustion, and irritability [31]. As the subjects in this study were a population with small social jet lag of 24.2 min during the control period, changes may be seen if a larger SJL population is targeted.

An analysis of defecation showed that the additional granola snacks increased the frequency of defecation by 1.2 per week. The fact that this effect was observed even in subjects who originally scored well in terms of the fecal characteristics and constipation scale is an indication of the usefulness of this form of breakfast. As shown in Figure 2D, the intervention in this study increased the breakfast adequacy rate of dietary fiber by an average of 26.4%, which is believed to have encouraged more defecation. Dietary fiber is known to have viscosity and water-retention properties [32], which increases the fecal bulk and lubricates the transport of stools in the bowel. The promotion of defecation by fiber-rich foods has been confirmed by a range of studies in adults [33,34]. There have been fewer reports in children [35], but the results of this study suggest that, in children as in adults, dietary fiber may promote defecation. The granola snacks used in the study were made mainly from oats, which contain soluble and insoluble fiber, which may have stimulated the production of short-chain fatty acids (SCFAs) [36]. SCFAs may promote peristalsis of the intestinal tract, shortening the time taken for stool to pass through the colon, and consequently facilitating defecation [37,38]. Furthermore, improvement of the intestinal environment may also have contributed to the results [39]. Ingesting soluble fiber-rich food in the morning rather than in the evening promotes lactic acid production, changes the composition of the intestinal microflora, and improves constipation [40]. In the future, it will be desirable to verify this improvement effect on subjects with defecation difficulties, and also to add changes in stool volume and intestinal microflora to the evaluation.

This study had a limitation in some factors (sample size, region, breakfast habit of subjects and research schedules). First, the required sample size was statistically calculated as 34, although only 26 subjects participated, and 21 were analyzed in this study. In the future, we intend to conduct the trial with larger number of subjects. Second, the study was conducted with elementary school children in a limited area of urban in Japan (Shinjuku, Tokyo). Therefore, it is necessary to recruit subjects from various regions to generalize these results. Third, all subjects in the present study were habitually consuming breakfast on a regular basis. Therefore, the effects of the breakfast intervention on skippers should be evaluated in the future. Fourth, in the current experiments, we conducted the first test trial for 1 week and the second test trial for 1 week with a 1 week washout. This schedule is enough to detect sleep change [41] and defecation change [40], but longer trial weeks and longer washout weeks may provide more positive data. Lastly, only breakfast content was recorded in this study, so we do not know whether the overall daily nutrient intake was changed during test periods and resulted in sleep and defecation being affected.

## 5. Conclusions

The results of this study indicate that additional granola snacks are useful for improving the nutritional balance of children’s breakfasts. Furthermore, it is suggested that increasing the amount of breakfast and improving the nutritional balance of breakfast can alter lifestyle habits, including sleep and defecation. These findings may be important for determining an appropriate breakfast during childhood, an important period of physical and mental development.

## Figures and Tables

**Figure 1 children-10-00779-f001:**
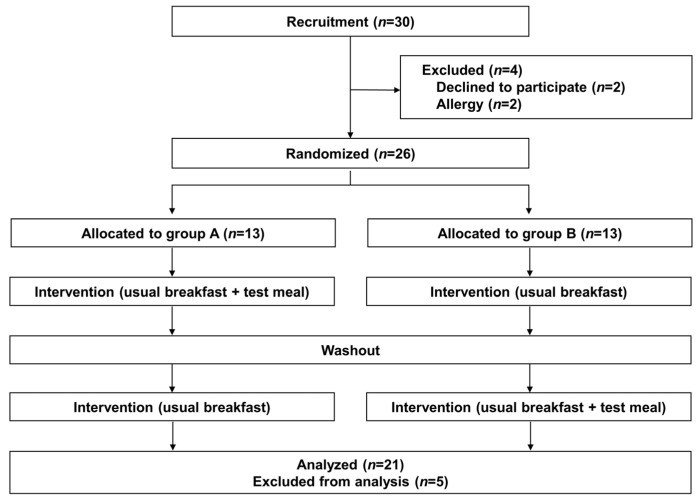
The workflow of this study. A total of 26 children participated and 21 of those were analyzed in this study.

**Figure 2 children-10-00779-f002:**
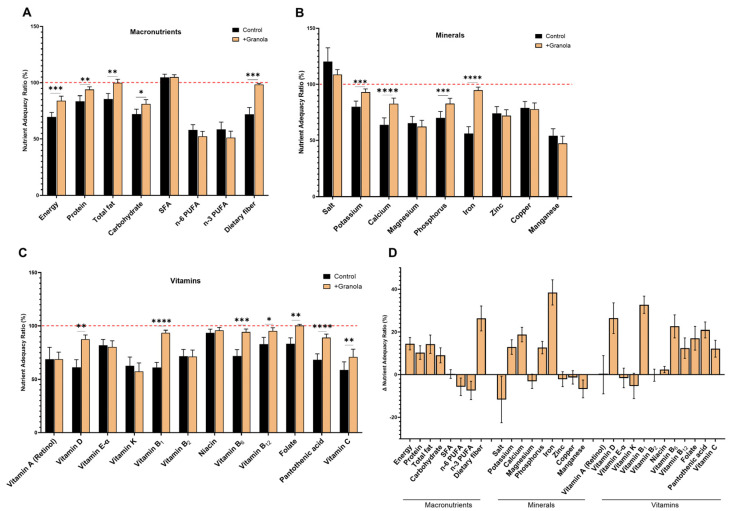
Calculated nutritional adequacy of (**A**) macronutrients, (**B**) minerals, and (**C**) vitamins in the control and intervention periods. Red dash line in the figure shows the 100% position. (**D**) Changes in nutritional adequacy rates during intervention. Mean ± SEM, *: *p* < 0.05, **: *p* < 0.01, ***: *p* < 0.001, ****: *p* < 0.0001.

**Figure 3 children-10-00779-f003:**
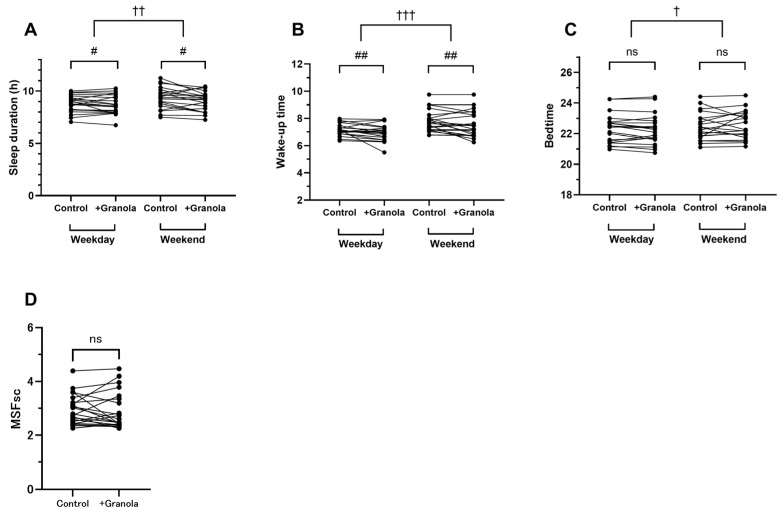
(**A**) Sleep duration on weekdays and weekends in the control and intervention periods. (**B**) Wake-up time on weekdays and weekends in the control and intervention periods. (**C**) Bedtime on weekdays and weekends in the control and intervention periods. (**D**) Sleep-corrected midpoint of sleep-in free days in the control and intervention periods. ^#,##^ Significant differences due to the intervention effect, *p* < 0.05, *p* < 0.01. ^†,††,†††^ Significant differences due to the day-of-the-week effect, *p* < 0.05, *p* < 0.01, *p* < 0.001. ns: not significant.

**Figure 4 children-10-00779-f004:**
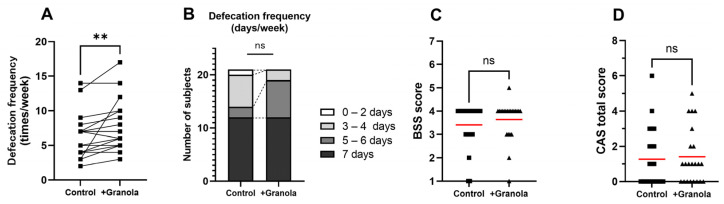
(**A**) Defecation frequency per week (times/week) in the control and intervention periods. (**B**) Distribution of days with defecation in the control and intervention periods. (**C**) Bristol stool scale scores in the control and intervention periods. (**D**) Constipation assessment scale scores in the control and intervention periods. Red dash: mean, **: *p* < 0.01, ns: not significant.

## Data Availability

The data presented in this study can be found in this published article and its Appendix A files.

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
