# Peer review of "Effects of Additional Granola in Children’s Breakfast on Nutritional Balance, Sleep and Defecation: An Open-Label Randomized Cross-Over Trial"

_children, 2023, doi:10.3390/children10050779_

Round 1

Reviewer 1 Report

Overall, a timely and well designed study dealing with an important health issue. Data collection, analysis, and presentation  are all appropriate. I have one question for the authors. With intervention adequacy of total fat increased & that of n-3 and n-6 PUFA decreased while that of SFA did not change. I wonder what contributed to this increase? was it MUFA which is not reported here.

Author Response

Overall, a timely and well designed study dealing with an important health issue. Data collection, analysis, and presentation are all appropriate. I have one question for the authors. With intervention adequacy of total fat increased & that of n-3 and n-6 PUFA decreased while that of SFA did not change. I wonder what contributed to this increase? was it MUFA which is not reported here.

We appreciate your comments on our study.

We consider as you do.

There was no statistically significant difference in the sufficiency rates of n-3, n-6 PUFA and SFAs, while the sufficiency rate of total lipids increased. Granola contains MUFA, which may be responsible for the increase in total lipids. However, we cannot be certain, as we did not assess the intake of MUFA in this study.

Please see lines 196-199.

Again, we appreciate you for giving us the opportunity to brush-up our manuscript with your valuable comments. We have worked hard to incorporate your feedback and hope that these revisions persuade you to accept our submission.

Reviewer 2 Report

Thank you for giving opportunity of reviewing the manuscript “Effects of Additional Granola in Children's Breakfast on Nutritional Balance, Sleep and Defecation”. As Japanese children decreases the ratio of eating breakfast, this investigation is useful to the improving that.

However, I have some concerns to publish this manuscript.

Introduction

l   I think introduction is well writing. Please add the concerns to skip breakfast for children (e.g. effect for study, activity, and performance).

l   Do you know the prevalence ratio of lack of breakfast in Japanese children?

Method

l   You recruited 30 children for investigation. How did you calculate the sample size at 26 subjects? 

l   When did you investigate this study? Please provide the date.

l   You wrote that the study consisted of one week in which subjects ate their usual breakfast and one week of eating the test food in addition to their usual breakfast, with a one-week washout period between the test periods. Is the period suitable for changing sleep and defecation? In addition, is the washout period sufficient? 

l   In washout period, how did you treat subjects who did not usually eat breakfast and who ate breakfast cereal?

l   How amount of breakfast cereal in investigation?

l   Are there subjects who usually eat breakfast cereal before investigation?

l   In general, significant level is set at 5% or 1%. Why you set at 10%?

Discussion

l   You provided three possibilities which caused shoter duration in weekends than weekdays. However, I concerned whether subjects stayed up late at weekend. It may solve the conceren when you provide the avarage time of going to bed.

l   You have to add the limitation section

l   Please provide CONSORT statement as a supplemental.

Author Response

Thank you for giving opportunity of reviewing the manuscript “Effects of Additional Granola in Children's Breakfast on Nutritional Balance, Sleep and Defecation”. As Japanese children decreases the ratio of eating breakfast, this investigation is useful to the improving that.

However, I have some concerns to publish this manuscript

Introduction

l   I think introduction is well writing. Please add the concerns to skip breakfast for children (e.g. effect for study, activity, and performance).

We appreciate your comments on our study.

It has been reported that school performance of breakfast-skipping children tends to be poorer than those who eat breakfast. We added the concern to introduction. Please see lines 42-43.

l   Do you know the prevalence ratio of lack of breakfast in Japanese children?

According to the 2019 National Health and Nutrition Survey in Japan [https://www.mhlw.go.jp/stf/seisakunitsuite/bunya/kenkou_iryou/kenkou/eiyou/r1-houkoku_00002.html], the breakfast-skipping rate for 7-14 year-olds in Japan is 4.4%.

Please see lines 41-42.

Method

l   You recruited 30 children for investigation. How did you calculate the sample size at 26 subjects?

The sample size was calculated using G*power. With an effect size of 0.5, an alpha error of 0.05 and a power of 0.8, a sample size was calculated as 34.

However, the number of subjects in this study was limited and did not meet the required sample size, so we consider increasing the number of subjects in the future. Please see lines 157-158 in the methods and lines 307-310 in the limitation section.

l   When did you investigate this study? Please provide the date.

the first intervention period: 31 Oct - 6 Nov 2022, washout: 7 - 13 Nov 2022, the second intervention period: 14 – 20 Nov 2022. As November 3rd was a holiday in Japan, sleep data on that day were not included in the analysis. Please see lines 120-122.

l   You wrote that the study consisted of one week in which subjects ate their usual breakfast and one week of eating the test food in addition to their usual breakfast, with a one-week washout period between the test periods. Is the period suitable for changing sleep and defecation? In addition, is the washout period sufficient? 

As prior studies with interventions similar to this study on children were scarce, we had to set our own study period. Regarding sleep, it has been widely reported that the diet of the day affects the quality of sleep1, and regarding defecation, it has been shown that Helianthus tuberosus (Dietary fiber accounts for 60%) consumption may change the defecation status and intestinal microflora within a week 2.

Based on these studies, we considered that sleep and defecation would change within a week, and set the study period and washout at one week.

1 St-Onge MP, Mikic A, Pietrolungo CE. Effects of Diet on Sleep Quality. Adv Nutr. 2016 Sep 15;7(5):938-49.

2 Kim, H.-K.; Chijiki, H.; Nanba, T.; Ozaki, M.; Sasaki, H.; Takahashi, M.; Shibata, S. Ingestion of Helianthus tuberosus at Breakfast Rather Than at Dinner is More Effective for Suppressing Glucose Levels and Improving the Intestinal Microbiota in Older Adults. Nutrients 202012, 3035.

We added this discussion in the limitation section. Please see lines 315-318.

l   In washout period, how did you treat subjects who did not usually eat breakfast and who ate breakfast cereal?

During the washout, we did not set any restrictions in this study.

Incidentally, none of the subjects in this study usually skipped breakfast. We asked participates to continue their usual breakfast habits during both two test periods and washout period. Please see lines 122-123.

l   How amount of breakfast cereal in investigation?

Added granola snacks were 26 g. The appearance was added to the supplemental (Figure S1). Please see supplemental data.

l   Are there subjects who usually eat breakfast cereal before investigation?

One of the participants had answered in the pre-questionnaire that he had a cereal eating habit. However, we confirmed that he did not consume cereal as usual breakfast during the study period. Please see lines 174-176.

l   In general, significant level is set at 5% or 1%. Why you set at 10%?

We apologize for our mistake.

We had set the significant tendency at 10%, but removed it as there was no relevant section.

Please see lines 167.

Discussion

l   You provided three possibilities which caused shorter duration in weekends than weekdays. However, I concerned whether subjects stayed up late at weekend. It may solve the conceren when you provide the avarage time of going to bed.

We may have misled you by our badly written description.

In this study, we show that the additional granola snacks lead to earlier wake-up time and shorter sleep time, and in the discussion part we mention the possibility that causes these changes.

It should be noted that the average bedtime on weekends was slightly later than on weekdays, but the intervention did not significantly change bedtimes. Therefore, we consider that the additional granola snacks had more effect on the rhythm of waking-up than of sleeping-in.

We show sleep parameters for each period in Table S4 in the supplemental.

Please see lines 254-278 and supplemental table S4.

l   You have to add the limitation section.

This study had limitation in some factors (sample size, region, breakfast habit of subjects and research schedules). First, the required sample size was statistically cal-culated as 34, however, only 26 subjects participated and 21 were analyzed in this study. In the future, we intend to conduct the trial with larger number of subjects. Second, the study was conducted with elementary school children in a limited area of urban in Japan (Shinjuku, Tokyo). Therefore, it is necessary to recruit subjects from various regions to generalize these results. Third, all subjects in the present study were habitually consuming breakfast on a regular basis. Therefore, the effects of the break-fast intervention on skippers should be evaluated in the future. Fourth, in the current experiments, we conducted first test trial for 1 week and second test trial for 1 week with 1 week washout. As this schedule is enough to detect sleep change1 and defecation change2, however, longer trial weeks and longer washout weeks may pro-vide more positive data. Lastly, only breakfast content was recorded in this study, so we do not know whether the overall daily nutrient intake was changed during test periods and resulted in affecting sleep and defecation.

Please see lines 307-320.

1 St-Onge MP, Mikic A, Pietrolungo CE. Effects of Diet on Sleep Quality. Adv Nutr. 2016 Sep 15;7(5):938-49.

2 Kim, H.-K.; Chijiki, H.; Nanba, T.; Ozaki, M.; Sasaki, H.; Takahashi, M.; Shibata, S. Ingestion of Helianthus tuberosus at Breakfast Rather Than at Dinner is More Effective for Suppressing Glucose Levels and Improving the Intestinal Microbiota in Older Adults. Nutrients 202012, 3035.

l   Please provide CONSORT statement as a supplemental.

We submit documentation on the CONSORT statement to the editor.

Again, we appreciate you for giving us the opportunity to brush-up our manuscript with your valuable comments. We have worked hard to incorporate your feedback and hope that these revisions persuade you to accept our submission.

Reviewer 3 Report

there's some enlightenment for children' breakfast in this manuscript. this topic is interesting for the improvement of dietary nutrients. I have three problems about this manuscript.

1. the sample size, how the sample size was calculated, I think the detailed information should be added.

2. how about the dietary nutrients and energy for one day. the results for the breakfast nutrients has been given, but which could not prove or clarify the nutrients intake for oneday become more.

3.the characteristic for the subjects were not given.

Author Response

there's some enlightenment for children' breakfast in this manuscript. this topic is interesting for the improvement of dietary nutrients. I have three problems about this manuscript.

  1. the sample size, how the sample size was calculated, I think the detailed information should be added.

We appreciate your comments on our study.

The sample size was calculated using G*power. With an effect size of 0.5, an alpha error of 0.05 and a power of 0.8, a sample size was calculated as 34.

However, the number of subjects in this study was limited and did not meet the required sample size, so we consider increasing the number of subjects in the future.

Please see lines 157-158 in the method section and lines 307-310 in the limitation section.

  1. how about the dietary nutrients and energy for one day. the results for the breakfast nutrients had been given, but which could not prove or clarify the nutrients intake for one day become more.

Only breakfast content was recorded in this study, so we do not know whether the overall daily nutrient intake become more. Please see lines 318-320 in the limitation section.

3.the characteristic for the subjects were not given.

Age, height and weight of subjects in Groups A and B were added to the supplemental data (Table S1). Both height and weight were close to the average for Japanese of that age.

Please see lines 106-108 and supplemental data (Table S1).

Again, we appreciate you for giving us the opportunity to brush-up our manuscript with your valuable comments. We have worked hard to incorporate your feedback and hope that these revisions persuade you to accept our submission.

Round 2

Reviewer 2 Report

Thank you for your emendations carefully.

Reviewer 3 Report

I agree to accept in present form in the systerm.